# Glycation-Associated Diabetic Nephropathy and the Role of Long Noncoding RNAs

**DOI:** 10.3390/biomedicines10102623

**Published:** 2022-10-19

**Authors:** Ankita Durge, Isha Sharma, Rashmi Santosh Tupe

**Affiliations:** 1Symbiosis School of Biological Sciences (SSBS), Symbiosis International (Deemed University), Lavale, Mulshi, Pune 412115, Maharashtra, India; 2Department of Medicine, Feinberg School of Medicine, Northwestern University, Chicago, IL 60611, USA

**Keywords:** diabetes, diabetic nephropathy, glycation, long noncoding RNA, biomarkers, therapeutics

## Abstract

The glycation of various biomolecules is the root cause of many pathological conditions associated with diabetic nephropathy and end-stage kidney disease. Glycation imbalances metabolism and increases renal cell injury. Numerous therapeutic measures have narrowed down the adverse effects of endogenous glycation, but efficient and potent measures are miles away. Recent advances in the identification and characterization of noncoding RNAs, especially the long noncoding RNAs (lncRNAs), have opened a mammon of new biology to explore the mitigations for glycation-associated diabetic nephropathy. Furthermore, tissue-specific distribution and condition-specific expression make lncRNA a promising key for second-generation therapeutic interventions. Though the techniques to identify and exemplify noncoding RNAs are rapidly evolving, the lncRNA study encounters multiple methodological constraints. This review will discuss lncRNAs and their possible involvement in glycation and advanced glycation end products (AGEs) signaling pathways. We further highlight the possible approaches for lncRNA-based therapeutics and their working mechanism for perturbing glycation and conclude our review with lncRNAs biology-related future opportunities.

## 1. Introduction

With the advancement in high-throughput sequencing technologies, including RNA sequencing and next-generation sequencing, researchers’ understanding of genome and transcriptome has improved remarkably. The advanced genome study started by several consortia such as the Human Genome Project, Encyclopedia of DNA element (ENCODE), and Functional Annotation of the mouse/mammalian genome (FANTOM) has eventually publicized that only a small portion (3%) of the complete genome relates to known protein-coding genes. The remaining amount (97%) is the so-called ‘unchartered territory of noncoding RNA’ with small fractions decoded [1]. The established understanding of gene regulation in nature revolves around the central dogma of life (DNA → mRNA → protein); however, the rediscovery of noncoding RNA (ncRNA) genes and their role in the regulation of protein-coding genes has stirred the world [2]. The advanced genomic studies pointed out that most RNA transcripts are noncoding and identified about 35,000 ncRNA transcripts exhibiting signatures such as mRNA, including polyadenylation, capping, and splicing [3].

Moreover, due to their mRNA-like features, these newly identified ncRNA transcripts are difficult to distinguish from protein-coding transcripts. Despite these cumbersome challenges, various phases of these consortia identified almost around 27,919 long noncoding RNAs (lncRNAs) [4]. These findings suggested that human genome transcription is ubiquitous and produces thousands of ncRNA transcripts [5]. The ncRNAs can be further grouped into two classes based on their nucleotide number. One group consists of RNA genes with fewer than 200 nucleotides (microRNAs, small interfering RNAs, Piwi-interacting RNAs, and small nucleolar RNAs) [6], while the other group comprises RNA genes with more than 200 nucleotides, called lncRNAs [7]. The lncRNAs are present abundantly in both animals and plants, and their presence in the plant kingdom can be seen as an example of combinatorial transcriptional regulation [8]. The expression patterns of lncRNAs depend on many biological factors, including high glucose (HG), free fatty acids, growth factors, and inflammatory cytokines; hence, they are considered biomarkers in many pathological conditions (Table 1) [9,10,11,12,13,14,15,16,17,18,19,20,21,22,23,24].

### 1.1. Structural Properties of LncRNAs

Many lncRNAs transcribe and process like mRNAs (Figure 1). The generative pathways (histone-modification patterns, splicing, spliceosome signals, and exons/introns segments) for lncRNAs are similar to those of protein-coding genes [25]. Nevertheless, unlike mRNAs, a significant portion of lncRNAs are predominantly located in the nucleus, then in the cytoplasm [26]. Additionally, lncRNAs are broadly divided into categories based on genomic locations, such as 1. Intergenic lncRNAs (located between two separate genes), 2. Intronic lncRNAs (located between two exons), 3. Sense, or 4. Antisense lncRNAs (located on the same or opposite strands of protein-coding transcripts) and 5. Bidirectional lncRNAs [27,28]. This location-based classification is helpful for structural recognitions and annotations; however, lncRNAs’ biological functions are equally essential for more accurate classification. The majority of the lncRNAs have a half-life of more than 16 h. The intergenic and *cis*-antisense lncRNAs are more stable than intronic lncRNAs.

### 1.2. Functional Properties of LncRNAs

#### 1.2.1. Role in Transcription Regulation

Gene regulation by lncRNAs can be facilitated through one of the two non-mutually exclusive ways: 1. By regulating adjacent gene expression and 2. By regulating chromatin state [29], thereby influencing neighboring gene expression. In both cases, lncRNA can either silences or enhances targeted gene expression [30]. The lncRNA Xist provides an ideal example of transcriptional silencing by spreading on a large portion of genes of one of the two X chromosomes [31]. LncRNAs Gas6 and SWINGN are well-studied examples of enhancer and enhancer-associated lncRNAs, respectively, to promote the transcriptional activation of targeted proteins [32].

#### 1.2.2. Role in Post-Transcriptional Regulation

Regulation of post-transcriptional events comprises targeted protein modifications and their subcellular localization. The process includes mRNA splicing, processing, and nuclear export, with the help of RNA binding proteins (RBP). LncRNAs regulate gene expression at post-transcriptional levels by binding to RBP via RNA sequence motifs. LncRNA-mediated post-transcriptional regulation increases or decreases targeted mRNA and/or protein expression and their nuclear trafficking [33]. Additionally, lncRNAs can alter mRNA stability and splicing in a context-dependent manner [34]. The mRNA splicing regulation by lncRNAs involves either forming lncRNA-RNA hybrids with pre-mRNA or modifying the availability of splicing factors.

#### 1.2.3. Role in Epigenetic Regulation

Many lncRNAs have been seen to regulate the epigenetic architecture of various proteins. These lncRNAs can either guide the epigenetic regulators to specific protein-coding loci or trigger the enhancer-promoter crosstalk. Their role as re-writers of chromatin folding (through chromatin modifying proteins such as methyltransferases and deacetylases) is documented in various studies [35]. Recent advances in detecting lncRNA-chromatin association have lightened their role at the epigenetic level [36]. Owing to their negatively charged surfaces, lncRNAs can neutralize positively charged histone proteins and de-compact the chromatin [37].

## 2. Techniques for Identification of LncRNAs

Often, lncRNAs are seen to express at low levels with reduced splicing efficiencies, making it difficult to isolate and identify them. Currently, there are three widely used methods to detect and identify lncRNAs, and these methods are non-mutually exclusive and, therefore, often used in combination within the same experiment.

### 2.1. Immunoprecipitation for LncRNAs Isolation

RNA immunoprecipitation (RIP) is used to isolate lncRNAs from a given RNA pool. The desired lncRNA can be crosslinked to cellular components prior to its isolation. Zhao and the group isolated and amplified lncRNA Xist using RIP and RT-PCR, respectively. Further, they discovered another lncRNA RepA crosslinked to lncRNA Xist, by using co-immunoprecipitation [38]. Other variants of RIP are protein-cross-link and immunoprecipitation and RNA-chromatin Immunoprecipitation [39]. The overall success rate of the assay is highly dependent on the affinity and specificity of the lncRNA-protein-antibody interaction. 

### 2.2. Microarray for Known LncRNAs Identification

Microarray-based techniques use genome-wide screening approaches to identify lncRNAs. Current microarray techniques can study expression profiles of lncRNAs interacting with mRNAs or miRNAs in various diseases [40]. Nevertheless, compared to RNA sequencing, the cost and simplicity of microarray assays make it the first choice in many applications.

### 2.3. RNA Sequencing for Novel LncRNAs Identification

A recently developed and powerful molecular biology tool is RNA sequencing. It is based on principles of next-generation sequencing to identify and quantify novel lncRNAs [41]. Though RNA seq is a time and cost-effective technology, it certainly has several unbeatable advantages, such as single nucleotide resolution, unbiased detection of previously unknown lncRNAs, and use of computational approaches for better understanding of lncRNA-mRNA-based co-expression studies [42].

## 3. Working Models of LncRNAs

Based on acting mechanisms, lncRNAs are classified into five archetypes/models: 1. Molecular signals, 2. Molecular decoys, 3. Enhancer RNAs, 4. Guides, and 5. Scaffolds (Figure 1). As some lncRNAs can act by single or multiple models, these archetypes are not mutually exclusive [43]. Transcriptions of some lncRNAs are time, external stimuli, and developmental stage-specific; such lncRNAs serve as ‘molecular signals’ and are transcribed in response to various environmental cues. Moreover, these lncRNAs either exhibit regulatory functions or are mere by-products of other transcriptions. Hence, their transcription is often considered a biomarker for a specific biological event. For example, lncRNAs HOX transcript antisense RNA (HOTAIR) and HOXA transcript at the distal tip (HOTTIP) signal anatomic position. In contrast, expressions of lncRNAs Promoter of CDKN1A antisense DNA damage activated RNA (PANDA) and Cold-Assisted Intronic noncoding RNA (COLDAIR) induced by DNA damage and external stimulus of cold, respectively [44,45]. Some lncRNAs can act as ‘molecular decoys’, regulating expressions of several mRNAs and proteins by limiting their transcription or translation. Such lncRNAs can sequester mRNAs and proteins in either nucleus (nuclear subdomains or chromatin) or cytoplasm. For example, lncRNA Telomeric Repeat containing RNA (TERRA) physically interacts and sequesters telomerase RNA template [46], whereas nuclear lncRNA Metastasis-Associated Lung Adenocarcinoma Transcript-1 (MALAT1) sequesters several serine/arginine splicing factors [47]. The third archetype includes lncRNAs that can ‘guide’ transcription machinery for the transcription of specific proteins in *cis* (protein transcribed from the neighboring gene) or *trans* (protein transcribed from the distantly located gene). For example, lncRNAs X-inactive specific transcript (Xist), COLDAIR, and HOTTIP can guide transcription of proteins in *a cis* manner [30], whereas lncRNAs HOTAIR and linc-p21 act in a *trans* fashion. LncRNAs in the fourth archetype serve as a structural platform for assembling multi-component complexes such as the ribonucleoprotein complex. For example, presence of lncRNA HOTAIR [48] and ANRIL [49] at the locus, helps in recruiting multiple protein complexes. Recently, a new class of lncRNAs has been described—enhancers or e-RNAs, produced by specific enhancers. The levels of enhancers and e-RNAs are directly correlated with the level of mRNA synthesis of the targeted gene. Hence, e-RNAs may promote specific mRNA synthesis.

## 4. Diabetic Nephropathy at a Glance

Diabetes mellitus (DM) is a group of metabolic disorders with insulin secretion deficiency, impaired insulin action, or both resulting in hyperglycemia. Its two main subtypes are Type I (a chronic condition with insulin deficiency) and Type II (impaired insulin action and resistance) [50]. Additionally, gestational diabetes mellitus and specific types of diabetes due to other causes such as neonatal diabetes, maturity-onset diabetes of the young and drug- or chemical-induced diabetes are also commonly observed [51]. Rapid urbanization, a sedentary lifestyle, and unhealthy eating habits can be blamed for a sudden increase in global diabetes prevalence. According to the International diabetes federation, around 451 million people are living with diabetes, which will increase to 693 million people by the year 2045 [52]. Compared to diabetes, diabetes-associated complications are largely responsible for increased mortality rates. Sustained and prolonged hyperglycemia causes generalized vascular damage, leading to organ-specific diabetic complications and impaired wound healing [53]. Uncontrolled/poorly controlled diabetes results in long-term damage and failure of various organs, including nerves, eyes, heart and kidneys.. Many creative management approaches have been developing over the past several decades to lessen these complications. These pathophysiological complications can also result from impaired lncRNA regulatory networks, as described above (Table 1).

DM is the main acquired factor responsible for the progression of not only diabetic kidney disease (DKD), but it also accounts for half of the end-stage kidney disease (ESKD) cases [54]. Diabetic nephropathy (DN) can be traditionally defined as urine albumin excretion, reduced renal function, and nephron numbers for reasons other than urinary tract infections and renal diseases. According to reports worldwide, almost 40% of patients with diabetes are prone to develop DN. As the disease progresses, the kidney becomes less efficient. Total kidney failure can be seen in the late stages of DN. It is also the leading cause of ESKD, with a prevalence of 80% of cases globally [55]. In addition, life expectancy shortens by 16 years in DM patients experiencing DN. Furthermore, the standardized mortality rate has reached 31.1% in patients suffering from DM and DKD. Hence, finding therapeutic measures for treating DN is mandatory [56]. 

Factors responsible for DN progression:

***1.****Glomerular hyperfiltration:* Glomerular hyperfiltration is one factor that worsens renal damage and typically takes place at the early stages of DN. Elevated glomerular filtration rate (eGFR) can be seen as an adaptation for reduced functional nephron mass. The decreased nephron number is further accompanied by imbalanced glomerular hyperfiltration [57]. Additionally, it triggers glomerular sclerosis and microalbuminuria, at initial stage of DN. The glomerular hyperfiltration in DN can be attributed to an elevated renin-angiotensin system, disturbed tubuloglomerular feedback, and impaired afferent-efferent arteriole resistance. Factors such as sorbitol, AGEs, and insulin-like growth factor 1 also contribute to glomerular hyperfiltration in DN.

***2.****Mitochondrial dysfunction, reactive oxidative stress, and hypoxia:* The structure and functions of mitochondria are highly susceptible to reactive oxygen species (ROS)-related damage. As mitochondria generate about 90% of energy for normal cellular functioning, mitochondrial dysfunction contributes in many diabetes-related complications, including DN [58]. Mitochondrial dysfunction directly relates to decreased ATP production and loss of renal function in DN. Furthermore, a decrease in mitochondrial biogenesis is also observed as the DN progresses. Therefore, ways to stimulate mitochondrial biogenesis need to be investigated. Renal hypoxia is also a key factor for DN progression, as confirmed by preclinical studies, wherein hypoxia triggers HIF-1α production and increases ROS production and pro-inflammatory cytokines such as TGFβ and TNF. The increase in ROS further triggers HIF-1α production and continues this vicious cycle. 

***3.****Renal anemia:* Another common but significant complication of CKD, as well as DN, is anemia, which weakens renal function as the disease progresses. Renal anemia can be considered a middle to late-stage event in DN [59]. The risk factors contributing to renal anemia generally include aging kidneys, insufficient RBC production, and AGEs-mediated RBC deformities. In various studies on DN and renal anemia, it was observed that renal anemia is an independent marker of ESKD and a silent contributor to renal function deterioration [60].

***4.****The interplay between HIF-1α and HIF-2α:* Renal damage in DN can also be attributed to increased levels of HIF-1α and decreased levels of HIF-2α. High levels of HIF-1α increase tubular ROS production, cortical oxygen consumption, and tubulointerstitial inflammation. It takes an active part in the activation of pro-inflammatory cytokines as well as pro-fibrotic genes. On the other hand, suppressed levels of HIF-2α mRNA in kidney fibroblasts indicate its role in renal function [61]. 

## 5. Glycation and AGEs at a Glance

Maillard reactions, as studied by L.C. Maillard in the early 1900s, were initially used to add texture, flavor, and palatability to food items [62]. These chemical reactions were widespread among food chemists for their beneficial properties. However, during the 1990s, investigators observed that Maillard-like reactions also transpire naturally, at a slower rate [63]. At a steady state, free amino groups of proteins and peptides react with reducing sugars without an enzyme or catalyst and form a highly fluctuating and unstable Schiff’s base. This Schiff’s base, an imine analog of aldehyde (aldimine), undergoes spontaneous rearrangements to form a comparatively stable ketoamine, known as the ‘Amadori product’ [64]. Such Amadori products and the unstable Schiff’s bases crosslink with other proteins or peptides through amino groups (lysine/arginine) to form glycated proteins. Both extracellular and intracellular proteins are susceptible to this glycation process. Eventually, researchers realized that these late-staged, complex Maillard reactions are highly involved in diabetes and related complications [65]. 

### 5.1. Complex Network of AGEs Formation

The network of non-enzymatic glycation reactions leading to AGEs formation is multifactorial. As illustrated in Figure 2, three major pathways for AGEs formation are (1) Hodge pathway (by auto-oxidation of Amadori products), (2) Namiki pathway (by rearrangement of Schiff’s base), and (3) Wolff pathway (by oxidation of reducing sugars such as glucose, fructose, and glyceraldehyde). Auto-oxidation of carbohydrates and fatty acids also results in glycation and AGEs formation. The increased levels of glycoxidation and lipoxidation enhance the production and accumulation of carbonyl species, thereby triggering carbonyl stress. Additionally, the AGEs reservoir depends on their endogenous and exogenous sources. The principal endogenous source is hyperglycemia, whereas exogenous sources can be attributed to oral intake of a high AGEs diet [66] and tobacco smoking [67]. Often, it is seen that exogenous AGEs contribute primarily to the total AGEs reservoir, then endogenous AGEs. Furthermore, ‘high AGEs formation is directly proportional to high glucose concentration’ is a common misinterpretation. 

### 5.2. AGEs and Their Receptors on Cells

AGEs activate various cell signaling pathways by binding to specific cell surface receptors as follows.

#### 5.2.1. RAGEs

RAGEs are the 35 KDa cell surface receptors first characterized in 1992 by Neeper and group [68] (Figure 3). These are classified into three types: 1. Full-length RAGE, 2. Cleaved RAGE, and 3. Endogenous secretory RAGE. Full-length RAGE is a multi-ligand receptor. It is also a member of the immunoglobulin family of cell surface receptors. All three types of RAGEs are expressed by various cells, including adipocytes, podocytes, fibroblast, monocytes/macrophages, T-lymphocytes, endothelial cells, neuronal cells, and dendritic cells. The AGEs-RAGE interaction pushes a cell on fetal edges mainly by four mechanisms: 1. By inducing oxidative and nitrosative stress [69], 2. By activating Jak/Stat signaling [70], 3. By activating the transcription factors [69,70], and 4. By increasing vascular calcification [71]. The AGEs-RAGE interaction activates NADPH oxidase, as discussed elsewhere [72]. 

#### 5.2.2. AGER (1, 2 and 3)

AGER plays a vital role in AGEs signaling and endocytosis of AGEs-modified proteins. AGER complex comprises three distinct members viz: AGEs R1, R2, and R3 (Figure 3). All three receptors are present on a wide range of cells, including adipocytes, T-lymphocytes, monocytes, fibroblasts, macrophages, endothelial cells, mesangial cells, and neuronal cells. After binding to AGEs, AGER1 activation reduces RAGE expression and AGEs-mediated oxidative stress-dependent signaling [73,74]. AGER1 overexpression drastically reduces AGEs-induced H_2_O_2_ production, MAPK/p44/p42 phosphorylation, and Ras activation [75]. AGER1 down-regulates AGEs-induced inflammatory response by suppressing NF-κB activity in mesangial cells. Moreover, AGER1 regulates NADPH oxidase-dependent oxidative stress by protein kinase C-δ (PKC-δ) suppression. A randomized study on T2DM patients showed decreased AGER1 and sirtuin1 (SIRT1) expression levels [76]. The down-regulation of SIRT1 triggers SIRT1-dependant NF-κB activation, which triggers pro-inflammatory responses and decreases longevity at cellular and organism levels [77,78]. AGER2 is a second member of the AGER complex, located in the cytosol and effectively phosphorylated by PKC [73]. Although its biological role is poorly understood, several studies indicated its possible role in intracellular receptor signaling and vesicular trafficking [79]. The last component of the AGER complex is a member of the lectin family called AGER3 (Galectin-3). Studies on macrophages show redistribution of galectin-3 on the cell surface upon incubation with AGEs, indicating its high binding affinity for AGEs ligand. The studies of AGER3/galectin-3 in immune response modulation, cell growth, migration, adhesion, and differentiation are also noteworthy [80,81].

### 5.3. Role of AGEs in DN Development and Progression

A high AGEs diet, as well as endogenous AGEs formation and accumulation, directly impact kidney function. AGEs damage the kidney and disturb renal functions by either glycating proteins essential for signaling or unsettling the molecular environment by AGEs-RAGE signaling. That is why the kidney is one of the primary targets for AGE-mediated cellular damage. With DN progression, AGEs excretion through the kidney decreases, and AGEs levels in circulation and accumulation in tissues increase [82]. The accumulation of AGEs increases intracellular calcium by activating Ca^2+^ channels, resulting in podocyte apoptosis. Podocyte apoptosis is considered a marker of DN development.

On the other hand, reports show that maladaptation due to AGEs-RAGE signaling is prominent in DN-related tubulointerstitial injury. The AGEs-RAGE interaction enhances myo-inositol oxygenase levels, increasing NF-κB expression, and pro-inflammatory cytokines and fibronectin levels [83]. As DN progresses, it leads to renal function impairment and albuminuria [84]. One of the first biological effects of AGEs formation was observed with glycated hemoglobin. Generally, small soluble glycated proteins, due to their shorter half-life, do not impart drastic effects on any metabolic pathway and are used as biomarkers for DM diagnosis [85].

Nevertheless, the problems occur when larger molecules with longer half-lives form glycated crosslinks. For example, collagenase with a longer half-life and a common glycation target contributes to many complications, including renal diseases and diabetes [86]. AGEs-collagenase can also crosslink with soluble proteins and add to diabetic complications via vascular stiffening. Furthermore, glycation of collagen IV-associated thickening of the glomerular basement membrane is heavily involved in the onset of DN [87]. Similarly, AGEs formation on DNA leads to genetic rearrangement and congenital malformations. Elevated levels of DNA glycation-induced nucleoside adducts can be seen in patients with DN [88]. Furthermore, AGEs can also modify the half-life of glycated proteins [89]. Like AGEs, advanced lipoxidation end products are a principal cause of many metabolic diseases. Lipid peroxidation and advanced lipoxidation end products are also involved in lipid-induced renal disorders [90]. 

Furthermore, reports based on in vitro and in vivo studies link AGEs with the bio-pathology of insulin resistance. Insulin resistance plays a key role in the development of DN. By interacting with pro-inflammatory signals, AGEs lead to apoptosis of nephrons, resulting in renal fibrosis and DN. AGEs are also responsible for inducing ER stresses and lead to DN development [91]. Moreover, AGEs-RAGE signaling triggers intracellular pathways, as discussed above, thereby inducing pathophysiological changes and altering renal functions [92]. A recent study reported adverse effects of AGEs on renal cell damage. The effects of glycated albumin on renal cells showed disturbed NRF2 signaling mediators. The study further reported that AGEs hamper functional properties of albumin such as binding capacity and antioxidant activity [93]. 

## 6. Understanding the Role of LncRNAs in AGEs-Related DN

Studies have shown that lncRNAs play a significant role in AGEs-mediated metabolic malfunctions [94]. So far, the involvement of lncRNAs-mediated AGEs-RAGE signaling in cancer, the immune system, and neurodegenerative diseases has been studied in detail [95]. However, few studies have reported the role of lncRNAs in glycation-associated diabetic complications [15,96,97,98,99] (Table 2). 

### 6.1. LncRNA-Mediated RAGE Gene Expression and Signaling

The lncRNAs such as H9, MIAT, and MEG3 significantly regulate the pathogenesis of various diabetic complications. Recently, the role of lncRNA Arid2-IR in AGEs-induced retinal endothelial cells was also observed. By binding to Smad3, LncRNA Arid2-IR regulated levels of oxidative stress, inflammation, and apoptosis [96]. Similarly, lncRNA HOTAIR regulated RAGE expression and inflammation in acute myocardium infarction-induced rat models [100]. Likewise, the delayed process of diabetic wound healing in AGEs-induced fibroblast was associated with lncRNA URIDS [15]. Overexpression and knockdown studies also confirmed the role of another lncRNA MVIH in the AGEs-RAGE signaling pathway responsible for tumor induction and progression in cancer [101]. In AGEs-induced endothelial cells, lncRNA MEG3 regulated cell viability, proliferation, and apoptosis through the lncRNA MEG3/MiR-93/p21 mediated pathway during the onset of diabetic vascular diseases [97]. In a cell-based study on hypoxia/reoxygenation-injured cardiomyocytes, lncRNA SNHG12 down-regulated RAGE, NF-κB expression, and pro-inflammatory responses [102]. In a different context, the lncRNA TP73-AS1-mediated RAGE-HMGB1 signaling pathway upregulated NF-κB expression and pro-inflammatory cytokine levels [12,103]. The above-discussed lncRNAs can be a promising biomarker to detect complications in AGEs-RAGE signaling-mediated metabolism.

### 6.2. LncRNAs That Regulate AGER Gene Expression and Signaling

Notably, the expression pattern of lncRNA AGER-1 correlates with AGER expression levels (r = 0.360, *p* = 2.15 × 10^−18^) possibly by binding and sponging miRNA-185. Animal-based studies confirmed these results with suppressed cell proliferation rate, migration, and colony-forming efficiency in nude mice [104]. In a different context, lncRNA AGER-1 induced cell cycle arrest and promoted apoptosis, thereby inhibiting cell proliferation and migration efficiency of tumor cells in several types of cancer [105]. Yet another study indicated the sponging effect of lncRNA AGER-1 for miR-182, an inhibitor of AGER1 [106]. Recently, a study showed that both AGER and its positive regulator lncRNA AGER-1 have the significant diagnostic potential for lung adenocarcinoma. Both AGER and lncRNA AGER-1 regulate apoptosis, cell migration, and antitumor responses [107]. In summary, lncRNA AGER-1 can be a promising agent against AGEs signaling in metabolic disorders.

## 7. Future Perspectives: LncRNAs as Biomarkers, Therapeutic Agents, and Therapeutic Targets

### 7.1. LncRNAs as Biomarkers

Traditional drugs, herbal medicines, and synthetic compounds generally target single proteins/genes; hence, they are constricted in curing complex multifactorial diseases such as DN. LncRNA-based studies are therefore crucial in focusing on better and safer options for treatments. The excellent ratio of protein: non-protein coding sequence in our genome made us think about the possible role of non-protein coding RNA—especially lncRNAs, in various pathophysiological conditions. Through studies, the fact that lncRNAs hold potential for a biomarker candidature is clear [108].

One of the remarkable features of lncRNAs is their diverse source of occurrence. LncRNAs are present in cell lines, exosomes, tissue types, peripheral blood, serum, tears, saliva, and urine, making sample collection in various patient-based studies comparatively less laborious. The occurrence or suppressed expression of specific lncRNAs in a normal and diseased condition is a characteristic of an excellent disease-specific biomarker. For instance, lncRNA HOTAIR is deregulated in cancers such as lung, gastric, colorectal, and prostate, but lncRNA prostate cancer antigen 3 is found only in prostate tissues [109]. As a result, lncRNA prostate cancer antigen 3 is used as a potent prostate cancer-specific biomarker. In this regard, the presence of lncRNAs, their nature, expression pattern, and function in the diabetic kidney as opposed to the normal, healthy kidney will help understand the DN’s molecular insights. 

Furthermore, we must remember that specificity is the key to use as a biomarker. Due to cell/tissue-type and disease condition specificity, lncRNA holds an outstanding therapeutic as well as preventive potential, which can be explored with a great deal of study in the future (Figure 4). For instance, lncRNAs such as E330013P06, upregulated in a high-fat diet plus streptozotocin (STZ)-induced T2DM mice, but not in STZ-induced T1DM mice, indicating its disease-specificity. Corroborating with these results, the human ortholog of lncRNA E330013P06 also shows upregulation when studied in T2DM patients. Likewise, the expression of the lncRNA Dnm3os is also expressed in macrophages from T1DM, T2DM, and high-fat diet-induced insulin resistance and in accelerated atherosclerosis. Thus, particular lncRNA is a biomarker in diabetes with accelerated atherosclerosis. There are many lncRNAs, which are either specifically expressed or suppressed in certain diseases [28,110]. Such lncRNAs activate or inhibit gene/protein functions upon expression, thereby regulating vital cellular processes such as metabolism and senescence. It also has been observed that some lncRNAs often play the role of molecular sponges against miRNAs. Such lncRNAs also, by working as anti-miRNA agents, diminish pathological conditions. Likewise, some lncRNAs (Lethe, Dnm3os) are expressed when treated with specific diabetogenic factors such as HG or palmitic acid [111]. 

A biomolecule’s stability is equally important to consider as a biomarker, especially for quantitative and qualitative analysis. LncRNAs are often underestimated as they are easily degradable and less stable than miRNAs. However, through their postmortem brain tissue studies, Kraus et al. proved that some lncRNAs are more stable than miRNAs [112]. Most lncRNAs show stability with a half-life of more than 16 h (intragenic and *cis*-antisense lncRNAs) [113]. The stability of lncRNAs depends on post-transcriptional modifications, sub-cellular localization, exosomal secretion, and overall function. Similarly, higher stability of some lncRNAs is confirmed by studies such as overnight incubation at room temperature and RNase A digestion, along with their presence in saliva and tears. In addition to stability, a biomarker must also show efficacy throughout the studies. LncRNAs show more desirable diagnostic effectiveness towards a particular disease than the present synthetic drugs. A meta-analysis elucidating the use of lncRNA as a biomarker for lung cancer showed its sustained diagnostic efficacy in identifying patients with lung cancer from healthy people [114]. Finally, high throughput detection methods and sensitivity of detection are two prominent features of any ideal biomarker. The current detection methods, as described above, have improved the detection of lncRNAs with precision.

### 7.2. LncRNAs as Therapeutic Agents

Owing to their functional properties, lncRNAs can be used as potential therapeutic agents in glycation-associated diabetic complications. The role of many lncRNAs in the pathology of diabetes-associated complications is revealed. Furthermore, their association with cellular processes such as inflammation, oxidative and ER stress, and mitochondrial dysfunction is now well-established. Identifying previously unknown lncRNAs involved in diabetic complications is now relatively easy because of rapid advances in RNA-sequencing technologies. Moreover, single-cell technology is valuable for detecting lncRNA expressed in different cell types. Additionally, a data library of lncRNAs expressed in cells and tissues of healthy individuals as well as diabetic patients is getting produced [4]. These data sets are instrumental in understanding many aspects such as an individual’s susceptibility towards diabetes, the pattern of regulation/deregulation of specific lncRNAs, and, more importantly, recognition of the excellent lncRNA (therapeutic agents). On the contrary, some lncRNAs act as ‘natural antisense transcripts,’ thereby naturally suppress the effectiveness of a given disease. Such lncRNAs can be used to design potential therapeutics (Figure 4). Likewise, synthetically designed small molecules such as NP-C86 can stabilize the expression of some lncRNAs and improve further cell signaling [115]. More research towards producing synthetic molecules and natural compounds, which are investigating increasing the stability of lncRNAs, is needed.

Recently, the use of exosomal ncRNA cargo as biomarkers and modulators of various diabetic complications is gaining more attention. Several studies have reported the role of miRNA exosomes as pathological modulators and biomarkers in T1DM and T2DM. The exosomal miRNAs such as miR-21-5p, miR-375-3p, and miR-133b are upregulated in diabetic patients and considered potent biomarkers, whereas miR-486, miR-222, and miR-146a are used as therapeutic agents [116]. Although studies on exosomal lncRNAs in diabetic complications are comparatively fewer, some lncRNAs such as p3134 showed better results when studied in serum exosomes. Further investigation showed lncRNA p3134 exosome-mediated upregulation of β-cell function preservation and insulin secretion in high glucose conditions [117]. Understanding the mechanism behind lncRNA exosome biogenesis, packaging, secretion, and uptake will help us improve our basic knowledge and design novel strategies to treat diabetic complications.

Interestingly, some studies have shown that few lncRNAs act as a double-edged sword in a similar diabetic condition, indicating their dual response in regulating the disease. According to some reports, expression of lncRNA Gas5 decreases in HG-induced renal cells and patients with DN. Its overexpression studies show a reduction in pyroptosis, oxidative stress, cell proliferation, and fibrosis by inhibiting miR-452-5p and miR-221 in a different context, indicating the Gas5-mediated downregulation of DN. However, some other studies show that lncRNA Gas5 increases in HG-treated renal cells and STZ/HFD-induced mouse models and upregulates renal fibrosis and cell apoptosis by downregulating miR-27a and miR-96-5p in different contexts, hence enhancing DN progression. Another lncRNA TUG1 decreases in lipopolysaccharide/HG-treated renal cells as well as in DM rats, according to some reports.

Additionally, its overexpression protects renal cells from inflammation and fibrosis by targeting miR-223 and miR-21 in different milieus, indicating its negative role in disease progression. Conversely, some reports showed its upregulation in hyperglycemia-induced podocytes and renal ischemia/reperfusion injury models. Further, silencing of TUG1 protects from ischemia/reperfusion-induced inflammation and apoptosis, decreasing the disease progression [118]. LncRNA MIAT also shows a similar type of regulation in DN. Using such lncRNAs as a biomarker or therapeutic agent is very promiscuous as they may or may not be the ideal indicator of a disease. More research is needed to identify and investigate the nature of such ambiguous lncRNAs.

### 7.3. LncRNAs as Therapeutic Targets

From various studies and experimental setups, it is clear that miRNAs can be easily targeted in vitro and in vivo. However, targeting lncRNAs is challenging owing to their large size, structural complexity, and low expression. Following are some techniques to target lncRNAs.

#### 7.3.1. RNA Gene Transcription Deregulation

Over the years, zinc finger nuclease has proved to be a site-specific manipulator of a genome. So far, zinc finger nucleases have created knockouts for protein-coding genes. However, their role in reducing noncoding gene expressions can also be explored [119]. Some RNA destabilizing elements can create a knockout-like effect on the lncRNA gene by working at a genomic level. RNA destabilizing elements are successfully used to reduce the expression of MALAT1 [120].

#### 7.3.2. Post-Transcriptional Deregulation

Post-transcriptional silencing of protein-coding mRNAs is not new to the molecular biology world. Similarly, lncRNAs can be targeted using specific siRNA associated with the RISK complex. The lncRNA–siRNA binding depends on target sequence-specific complementarity. For instance, in an animal model, the lentivirus-based shRNA knockdown expression of lncRNA Kcnq1ot1and improves cardiac functions and fibrosis [121]. Therapeutic stratagems based on artificially designed antisense oligonucleotides (ASO) have recently gained much attention. ASOs can also be designed to suppress the expression and functions of lncRNAs. ASO is a single-stranded oligomer that offers target-specific complementarity. Once bound, it can also degrade lncRNA by RNase H-mediated signaling. As far as lncRNA knockdown is concerned, ASO shows more target specificity and efficiency than siRNA [122]. However, the cellular uptake, off-target effect, and half-life of ASO are some factors that should be considered while designing the antisense oligomer [123]. Single-stranded oligonucleotides with DNA segments flanked by ‘locked nucleic acid’ can also be used to target sequence-specific lncRNA. The method also uses an RNAi-based knockdown of lncRNAs, just like ASO. Nevertheless, it is currently limited to oncogenic lncRNAs, which are explicitly localized to the nucleus. Similarly, lncRNA-associated in diabetic complications can also be targeted. Another technique that can be effectively used against suppressing lncRNA is CRISPR/Cas9 [124]. Moreover, these knockdown strategies may vary in ‘lncRNA-targeting efficiency’ based on the cellular localization of lncRNAs. It has been found that ASO suppresses nuclear lncRNAs, while RNAi methods are more effective against cytoplasmic lncRNAs. However, both methods can effectively suppress the dually expressing lncRNAs [125].

#### 7.3.3. Functional Disruption

Aptamers are chemically synthesized single-stranded oligonucleotides. They can bind and form tertiary structures with targeted lncRNAs. The aptamer may prove efficient in sequence-specific binding over siRNA or ASO. Furthermore, the chemical structures of aptamer can be easily modified to improve their half-life and stability. Wang et al. designed a chimeric aptamer, EGFR-coupled siHOTAIR, recognizing both EGFR and HOTAIR. Once recognized, this aptamer can knock down HOTAIR with coupled siRNA against HOTAIR [126]. Similarly, small chemically synthesized molecules with potential inhibitory properties could also prove helpful. Such molecules can be screened for their competitive binding to the binding site on lncRNA [127].

## 8. Conclusions

We are still falling behind in finding satisfactory treatments for DN. The disease continues to spread and increase mortality with each passing year. There is an urgent need to find pivotal and safer options to tackle this malady. Our present knowledge of lncRNAs and their involvement in AGEs-mediated cell signaling provides the basis and justification for considering them as potential markers in understanding glycation-associated DN pathogenesis. Further investigations into their interactions with biomolecules such as DNA, RNA, ncRNAs, and proteins in glycation-induced conditions may provide ways to utilize them as therapeutic agents or targets. Therefore, adding these RNA genes to our current armamentarium will prove beneficial for developing newer and better approaches to target DN and associated complications. 

## Figures and Tables

**Figure 1 biomedicines-10-02623-f001:**
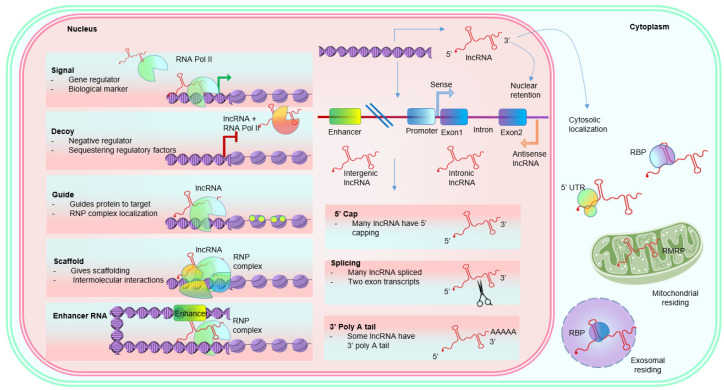
Archetypes and characteristics of lncRNAs along with their nuclear or cytoplasmic localizations. The various archetypes explain their mechanism of action in a non-mutually exclusive fashion. LncRNAs exhibit post-transcriptional signatures such as mRNA. Depending upon the splicing, lncRNA can either retain in the nucleus or localize in the cytosol. Abbreviations: LncRNAs, long noncoding RNAs; RNP, ribonucleoprotein; UTR, untranslated region; RMRP, RNA component of mitochondrial RNA-processing endoribonuclease; RBP, RNA-binding protein.

**Figure 2 biomedicines-10-02623-f002:**
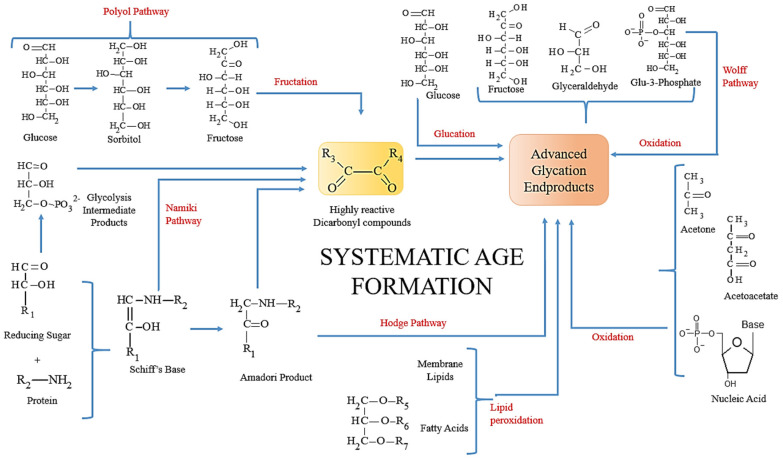
A complex network of AGEs formation: AGEs are formed through various pathways under oxidative and carbonyl stress conditions.

**Figure 3 biomedicines-10-02623-f003:**
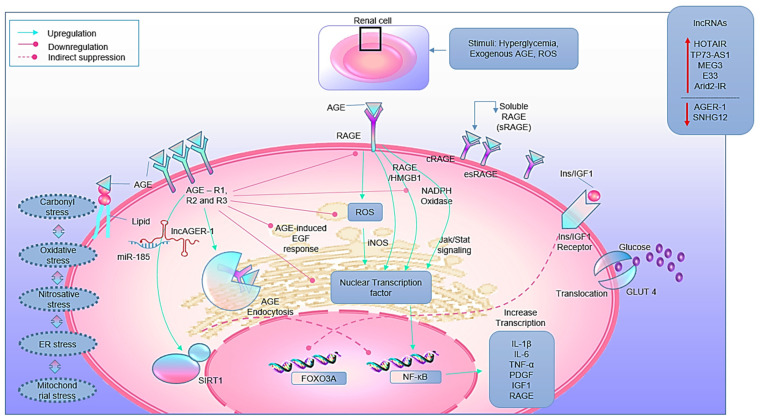
AGEs cellular signaling pathway through its receptors. AGE-RAGE interaction promotes NADPH oxidase and ROS production as well as regulates many downstream signaling pathways. AGER complex helps in the detoxification and degradation of AGEs. Abbreviations: IGF-1, Insulin-like growth factor-1; PDGF, platelet-derived growth factor; ISRE, interferon-stimulated response element; ROS, reactive oxygen species; SIRT1, silent mating type information regulator 2 homolog 1.

**Figure 4 biomedicines-10-02623-f004:**
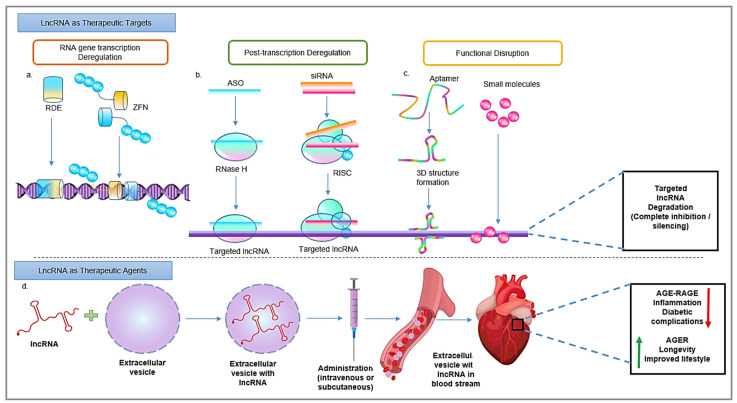
Potential application of lncRNAs as the future therapeutic agents or targets. LncRNAs can be targeted if they have an active role in disease progression or can be used in a therapeutic measure. Abbreviations: RDE, RNA destabilizing elements; ZFN, zinc finger nuclease; ASO, antisense oligonucleotide.

**Table 1 biomedicines-10-02623-t001:** LncRNAs involved in various diabetes-associated complications.

Diabetic Complications	lncRNA Involved	Mode of Action	References
Diabetic neuropathy	lncRNA NEAT1	Regulate disease progression by targeting two miRNAs, miR-183-5p and miR-433-3p.	[9]
	lncRNA TP73-AS1	Sponges decreases miR-142 and upregulates HMGB1 expression as well as promotes cell proliferation. Its silencing decreases neuropathic pain.	[10]
Diabetic retinopathy	lncRNA HOTTIP	Induces p38/MAPK signaling and promotes retinal cell inflammatory response and diabetic retinopathy progression.	[11]
	lncRNA BANCR	Regulates cell apoptosis	[12]
	lncRNA MEG3	Regulates VEGF and TGF-β1 expressions	[13]
Inflammatory diabetes complications	lncRNA DRAIR	Regulates macrophages/monocyte pro/anti-inflammatory phenotypes in T2DM. Down-regulated by diabetogenic factors	[14]
Diabetic wound healing	lncRNA URIDS	Impairs collagen production and crosslinking by interacting with Plod1 and delays wound healing	[15]
	lncRNA MALAT1	Increases wound healing and upregulate fibroblast activation in diabetic mice by activating the HIF-1α signaling pathway.	[16]
Diabetic cardiomyopathy	lncRNA HOTAIR	Downregulates DCM effects by activating SIRT1 expressions and sponging miR-34a.	[17]
	lncRNA Kcnq1ot1	Promotes pyroptosis by regulating expressions of miR-214-3p and caspase-1	[18]
	lncRNA H19	Regulates high glucose-induced apoptosis by targeting VDAC1. It also improves left ventricular function when overexpressed.	[19]
	lncRNA Crende	Negatively regulates cardiac fibroblast differentiation. Also, its expression is induced by Smad3 in cardiac fibroblasts. It inhibits myofibroblastic gene transcription.	[20]
	lncRNA TUG1	Its knockdown lessened DCM-induced hypertrophy and diastolic function. Also, its silencing upregulates the expression of some miRNAs	[21]
Diabetic nephropathy	lncRNA Blnc1	Attenuates renal fibrosis and inflammation and affects oxidative stress by NF-κB and NRF2/HO-1 pathways	[22]
	lncRNA TCF7	It acts as a sponge against miR-200c and triggers endoplasmic reticulum stress in patients with DN.	[23]
	lncRNA Gas5	Alleviates cell proliferation and fibrosis sponging miR-221 and upregulates SIRT1	[24]

Abbreviations: lncRNA MEG3, maternally expressed gene 3;, lncRNA Neat1, nuclear paraspeckle assembly target 1; HMGB1, High mobility group box 1; lncRNA URIDS, lncRNA Upregulated in Diabetic Skin; MAPK, Mitogen-activated protein kinase; lncRNA MALAT1, Metastasis-Associated Lung Adenocarcinoma Transcript 1; HIF-1α, Hypoxia-inducing factor - 1α; lncRNA Gas5, Growth Arrest-specific 5; lncRNA DRAIR, diabetes regulated anti-inflammatory RNA; lncRNA Crende, colorectal neoplasia differentially expressed; Smad3, SMAD family member 3lncRNA TUG1, taurine up-regulated gene 1lncRNA H19, Imprinted maternally expressed transcript; TCF7, Transcription factor 7; lncRNA BANCR, B-Raf proto-oncogene serine/threonine kinase-activated non-protein coding RNA; VEGF, vascular endothelial growth factor; SIRT1, silent mating type information regulator 2 homolog 1; DCM, diabetic cardiomyopathy; T2DM, type 2 diabetes mellitus.

**Table 2 biomedicines-10-02623-t002:** LncRNAs involved in glycation/AGE-associated diabetic complications.

LncRNA Involved	Diabetic Complication	Mode of Action	References
lncRNA Arid2-IR	Diabetic retinopathy	Regulates oxidative stress, inflammatory responses, and endothelial cell dysfunction via interacting with Smad3	[96]
lncRNA MIAT	Diabetic retinopathy	AGE-induced HRPCs MIAT and CASP1 expressions increase, followed by the release of IL-1β, IL-18, and suppression of cell viability.	[98]
lncRNA MEG3	Diabetic vascular diseases	Upregulates in AGE-induced cells and suppresses cell viability and proliferation by modulating the MEG3/miR-193/p21 pathway	[97]
lncRNA URIDS	Diabetic Wound Healing	Upregulates upon AGE induction. Regulates collagen production and deposition by targeting Plod1. It delays the wound healing process.	[15]
lncRNA E330013P06	Inflammatory diabetes complications	Increases inflammatory response upon AGE induction; by triggering pro-inflammatory gene. It also enhances foam cell formation.	[99]

Abbreviations: lncRNA MEG3, maternally expressed gene 3; lncRNA URIDS, lncRNA Upregulated in Diabetic Skin; lncRNA MIAT, myocardial infarction-associated transcript; HRPCs, Human retinal pericytes.

## Data Availability

Not applicable.

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
