# Peer review of "Glycation-Associated Diabetic Nephropathy and the Role of Long Noncoding RNAs"

_biomedicines, 2022, doi:10.3390/biomedicines10102623_

Round 1
Reviewer 1 Report
Regarding the present manuscript, the following issues could be mentioned:
1. The reference regarding diabetes classification should be replaced with a newer reference.
2. Other types of diabetes include many other types than the ones mentioned in lines 149-150, so-called specific diabetes. example of reference: https://diabetesjournals.org/care/article/45/Supplement_1/S17/138925/2-Classification-and-Diagnosis-of-Diabetes
3. A more recent reference can be cited for the description of diabetes complications: 48.
4. Some references must be revised: 10, 28, 45, 60.
Reviewer 2 Report
The topic of this article is novel and has certain practical significance. It is worth mentioning that the article drawing is more exquisite, reflecting the author 's certain drawing and writing ability. Based on the treatment of IncRNA and its interference with the signal pathway of advanced glycation end products, this paper expounds its biological and therapeutic effects on DN. It has some enlightening significance for the treatment and mechanism of DN.
However, there are some problems with the thesis, which are summarized as follows.
1. In ' 1.2.1.role in transcription regulation ', ' the two non-mutually exclusive ways : ' it is recommended to use 1. and 2. for more clarity and to add references.
2. After the first sentence of ' 1.2.2.role in post-transcriptional regulation ' and ' 1.2.3.role in epigenetic regulation ', it is suggested to expand the explanation appropriately.
3. In ' 4. Diabetic nephropathy at a glance ', it is recommended to remove the title ' 4.1.Factors responsible for DN progression '. Change to ' Factors responsible for DN progression : ( 1 ) Glomerular hyperfiltration : ( 2 ) Mitochondrial dysfunction, reactive oxidative stress, and hypoxia.... And so on.
4. It is recommended to appropriately reduce the content of ' 5.1.Complex network of AGEs formation ', ' 5.3.AGEs and their receptors on cells ', while appropriately focusing on ' 5.4.Role of AGEs in DN development and progression '.
5. From the full text, the core content of the article ' 6.Understanding the role of lncRNAs in AGEs-related DN ' occupies less space, and it is suggested to enrich the relevant expression and research.
6. ' 6.2 LncRNAs that regulate AGER gene expression and signaling ' cited literature is less, it is recommended to increase the cited literature to improve the persuasiveness.
7. In ' 7.3.LncRNAs as therapeutic targets : ' it is better if the therapeutic effects of LncRNAs are combined with DN.
